# Enhancing Multi-Tip Artifact Detection in STM Images Using Fourier Transform and Vision Transformers

**Tommaso Rodani** [1 2]  **Alessio Ansuini** [1]  **Alberto Cazzaniga** [1]

## Abstract

We address the issue of multi-tip artifacts in Scanning Tunneling Microscopy (STM) images by applying the fast Fourier transform (FFT) as a feature engineering method. We fine-tune various neural network architectures using a synthetic dataset, including Vision Transformers (ViT). The FFT-based preprocessing significantly improves the performance of ViT models compared to using only the grayscale channel. Ablation experiments highlight the optimal conditions for synthetic dataset generation. Unlike traditional methods that are challenging to implement for large datasets and used offline, our method enables on-the-fly classification at scale. Our findings demonstrate the efficacy of combining the Fourier transform with deep learning for enhanced artifact detection in STM images, contributing to more accurate analysis in material science research.

## 1. Introduction

Scanning Tunneling Microscopy is a powerful technique used to obtain high-resolution images of surfaces at the atomic level[1]. STM images are generated by scanning a sharp metal tip very close to the sample surface and measuring the tunneling current flowing between the tip and the sample. This current varies with the distance between the tip and the sample, allowing for precise mapping of the surface's topography.

During STM imaging, artifacts or distortions can arise that are not present on the sample surface. These artifacts often result from unpredictable surface interactions that alter the tip's shape, leading to highly nonlinear changes in the acquired data [2]. One prominent type of artifact is the duplication of sample structures, which occurs when the probe shape includes multiple atoms at the apex. This multi-tip or ghosting artifact results in duplicated signals, which complicates data interpretation[3].

Deep learning methods are widely used for image classification, with Convolutional Neural Networks (CNNs) traditionally serving as the foundation for computer vision algorithms. Recent advancements, such as Vision Transformer[4], have established Transformers as a dominant force in visual modeling due to their attention mechanism and large receptive field, which are crucial for visual tasks.[5],[6], [7]. While ViT models have achieved good performance, they typically rely on image features from the spatial domain[8]. Initial applications of machine learning to STM-related issues have been documented, enabling automated identification of defects[9], texture segmentaion [10] as well as denoising methods[11]. In addition, advancements have been made toward autonomous STM operation through ML-based tip shaping and lithography [12],[13],[14]. Despite these advancements, the multi-tip artifact issue in STM images has received limited attention, highlighting the need for further research in this area.

Previous studies link probe morphology and image quality through analytical simulations [15] or use inverse imaging of the probe via sample features [16],[17],[18]. While effective, these approaches are challenging to implement for general use, especially with large datasets, and existing software tools such as Gwyddion [19] are typically used offline after data acquisition. In contrast, our method enables on-the-fly classification at scale.

The Fourier transform, a powerful tool for extracting subtle information, can reveal features such as perturbations that are not noticeable in the spatial domain but become prominent in the Fourier domain[20]. In this paper, we address multi-tip artifacts in STM images by applying the Fourier transform to decompose image content into constituent frequencies. This transformation enhances the model's ability to identify and classify these artifacts accurately by making key information more discernible in the frequency domain.

[1]AREA Science Park [2]Università degli Studi di Trieste. Correspondence to: Tommaso Rodani <tommaso.rodani@areasciencepark.com>.

*Accepted at the 1st Machine Learning for Life and Material Sciences Workshop at ICML 2024.* Copyright 2024 by the author(s).

## 2. Method

We generated a synthetic dataset using experimental STM images originally recorded using an Omicron Variable Temperature STM (VT-STM) microscope. The dataset is composed of 2080 grayscale images of 400x400 pixels, manually labeled as either containing (82 images) or not containing (1998 images) the multi-tip artifact. We'll refer to artifact-free images as clean during the rest of the paper. To create a balanced synthetic dataset, we transformed a subset of clean images into synthetic multi-tip images. This was achieved by summing the clean image and N of its translations, where N represents the number of tips. The hyperparameters used for the dataset generation included the X and Y components of the translation vector set between 2 to 8 pixels, the intensity of the translation set between 50% to 80% of the original signal, and a uniform number of tips up to 12.

The images were then cropped to 224x224, a suitable size for training. The Fourier Transform produces a complex-valued output image, which can be displayed using either the real and imaginary parts or the amplitude and phase. We applied FFT to obtain the amplitude and phase, and saved each image as a three-channel image comprising greyscale, amplitude , and phase. Figure 1 below shows an example of a three-channel image before and after the application of the multi-tip artifact.

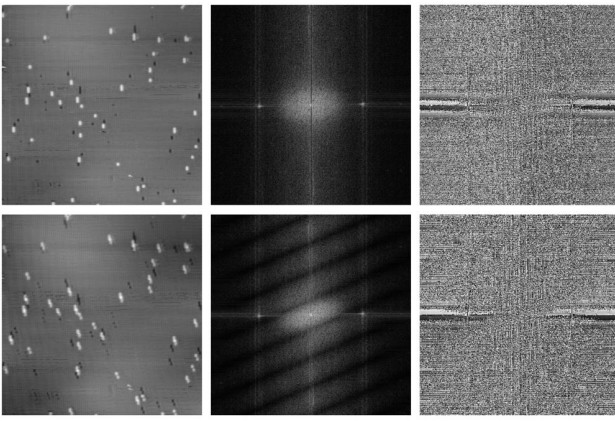

*Figure 1.* Synthetic multi-tip image
The three columns show the image channels: grayscale and the amplitude and phase of the FFT transformation. The first row corresponds to a clean image. The second row shows the same image after the application of the synthetic multi-tip artifact.

The dataset was split into 85% for training and 15% for testing. The training set was further divided into 90% for training and 10% for validation. All the experimental images with the multi-tip artifact were used exclusively in the test set.

We employed various pre-trained neural network architectures to fine-tune on the synthetic dataset, specifically ResNet with 18 and 50 layers, and Vision Transformer Base (ViT-B) with 16 and 32 layers.

The neural networks were trained using 50 epochs with early stopping, utilizing cross-entropy loss as the loss function and the SGD optimizer with a learning rate of 0.01 and momentum of 0.9. Data augmentation was applied through torchvision's TrivialAugmentWide.

## 3. Results

The results of the comparison on the test set are reported in Table 1, with the best results highlighted in bold. There is a considerable performance improvement in the ViT-B16 and ViT-B32 models using the proposed FFT-based preprocessing method compared to the classical use of only the grayscale channel. The improvement of ViT over ResNet can be attributed to ResNet's vulnerability to high-frequency noise, whereas ViT shows robustness. Multi-Head Self-Attention (MSA) in ViT performs low-pass filtering, highlighting the FFT-based method's advantage in preserving high-frequency components.

To better understand performance improvement, we tested each neural network using each component of our method: grayscale image, amplitude, and phase. The results of this experiment are shown in Table 2, where it is clear that the amplitude component is the main contributor to the increased performance. This was expected, as it is known that the amplitude holds more geometrical structure of features in the image.

To further explore the validity of our method, we carried out ablation experiments on the dataset. In Table 3, we verify the impact of different numbers of tips in the synthetic dataset generation. Using our FFT-based method, the number of tips does not affect performance. In contrast, with ViT-B16, the best model on the grayscale image, increasing the number of tips beyond 12 does not benefit the model. We also studied the effect of increased ranges of translation vectors in the synthetic dataset generation. As shown in Table 4, using a wider range has a detrimental effect on the model performance.

## 4. Conclusion

In this study, we address the challenge of multi-tip artifacts in STM images by leveraging the Fourier transform to decompose image content into constituent frequencies.

Our experimental results demonstrated that Vision Transformer models showed considerable performance improvements when using the proposed FFT-based preprocessing method compared to using only the greyscale channel. Fur-

*Table 1.* Classification accuracies of different models with and without our FFT-method on the test set.

| MODEL | ACCURACY | FFT-BASED |
|---|---|---|
| RESNET18 | 52.43 | × |
| RESNET18 | 57.07 | √ |
| RESNET50 | 58.84 | × |
| RESNET50 | 65.24 | √ |
| VIT-B/32 | 78.96 | × |
| VIT-B/16 | 88.11 | × |
| VIT-B/16 | 97.25 | √ |
| **VIT-B/32** | **97.86** | √ |

*Table 2.* Classification accuracies of different models for each component of our FFT-method on the test set.

| MODEL | IMAGE | AMPLITUDE | PHASE |
|---|---|---|---|
| RESNET18 | 52.43 | **60.06** | 53.04 |
| RESNET50 | 58.84 | **62.50** | 51.21 |
| VIT-B/16 | 88.11 | **97.56** | 66.15 |
| VIT-B/32 | 78.96 | **97.86** | 68.90 |

*Table 3.* Classification accuracies of ViT-B/16 on the test set using different range of possible tips on the synthetic dataset generation.

| TIPS | ACCURACY |
|---|---|
| 4 | 75.30 |
| 8 | 75.61 |
| **12** | **84.76** |
| 16 | 71.34 |

*Table 4.* Classification accuracies on the test set using different pixel ranges of translation vector on the synthetic dataset generation.

| PIXEL RANGE | ACCURACY |
|---|---|
| **2-8** | **97.86** |
| 10-30 | 85.97 |
| 10-50 | 92.98 |

ther analysis revealed that the amplitude component was the primary contributor to this improvement, underscoring its importance in capturing the geometrical structure of features in the images. Moreover, theoretical findings suggest that Multi-Head Self-Attention (MSA) inherently performs low-pass filtering on image signals, leading to rank collapse and patch uniformity issues in deep Vision Transformers [21]. This might explain why the FFT-based method outperforms, as FFT-based preprocessing transforms image data into the frequency domain, effectively highlighting and preserving high-frequency components. These components are crucial for detecting subtle features and structures in STM images that MSA's low-pass filtering might otherwise

smooth out. Additionally, ablation experiments provided valuable insights into the optimal conditions for synthetic dataset generation of multi-tip artifacts.

Overall, our study demonstrates the potential of combining traditional image processing techniques like Fourier transform with deep learning models for artifact detection in STM image analysis. Our method enables on-the-fly classification, which is currently done manually or offline. In future studies, we will include a baseline comparison with existing methods and expert practitioners. This approach not only enhances multi-tip detection but also facilitates more comprehensive analysis of microscopy images, advancing research in material science.

## Software and Data

The data that support the findings of this study are available from the corresponding author, upon reasonable request.

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
