# OpenReview forum: "Enhancing Multi-Tip Artifact Detection in STM Images Using Fourier Transform and Vision Transformers"
_ICML.cc/2024/Workshop/ML4LMS — ML4LMS Poster_

### Official Review · Reviewer_BRC5 · 2024-06-09
**A vision transformer based classifier to detect multi-tip artifacts in STM**

**Rating:** 6
**Confidence:** 5

**Review:**

In this paper, the authors employed a vision transformer to classify whether a given STM image has a multi-tip artifact. The model addresses a practical problem in the SFM measurements using deep learning models. Several innovations are introduced to improve model performance.

Strengths:
- The multi-tip data argumentation strategy provides a simple way to argument data and improved performance.
- The fourier features introduced are shown to improve performance.
- Detailed ablation study is provided to identify how different components are contributing to the final performance.

Weaknesses:
- No baseline for non-ML based models. Multi-tip artifacts are easy to spot by practitioners & existing software. The paper should be stronger if it is compared to a non-ML baseline to demonstrate the advantage of ML-based methods.

Suggestions for improvement:
- In Table 1, it is surprising that ViT has such a large improvement over ResNet. Do the authors have any explanations?
- It would be helpful if the authors can contextualize the performance of their model from the perspectives of a practitioner. How does the model compare with existing solutions of detecting multi-tip artifacts? How would a practitioner use the model in their workflow?

---

### Official Review · Reviewer_raTf · 2024-06-12
**Use of FFT with ViT and other models**

**Rating:** 8
**Confidence:** 5

**Review:**

Interesting use of FFT with different DL architectures. I would suggest of increasing the number of images in the dataset because ~ 2080 images is low for training a ViT and generalizing it.